# Emerging Advanced Technologies to Mitigate the Impact of Climate Change in Africa

**DOI:** 10.3390/plants9030381

**Published:** 2020-03-19

**Authors:** Priscilla Francisco Ribeiro, Anyela Valentina Camargo Rodriguez

**Affiliations:** 1CSIR—Crops Research Institute, Cereals Division, Fumesua, Kumasi AK000-AK911, Ghana; prisboat@yahoo.com; 2The John Bingham Laboratory, NIAB, 93 Lawrence Weaver Road, Cambridge CB3 0LE, UK

**Keywords:** climate change, bioinformatics, Africa, OMICS

## Abstract

Agriculture remains critical to Africa’s socioeconomic development, employing 65% of the work force and contributing 32% of GDP (Gross Domestic Product). Low productivity, which characterises food production in many Africa countries, remains a major concern. Compounded by the effects of climate change and lack of technical expertise, recent reports suggest that the impacts of climate change on agriculture and food systems in African countries may have further-reaching consequences than previously anticipated. Thus, it has become imperative that African scientists and farmers adopt new technologies which facilitate their research and provide smart agricultural solutions to mitigating current and future climate change-related challenges. Advanced technologies have been developed across the globe to facilitate adaptation to climate change in the agriculture sector. Clustered regularly interspaced short palindromic repeats (CRISPR)-CRISPR-associated protein 9 (Cas9), synthetic biology, and genomic selection, among others, constitute examples of some of these technologies. In this work, emerging advanced technologies with the potential to effectively mitigate climate change in Africa are reviewed. The authors show how these technologies can be utilised to enhance knowledge discovery for increased production in a climate change-impacted environment. We conclude that the application of these technologies could empower African scientists to explore agricultural strategies more resilient to the effects of climate change. Additionally, we conclude that support for African scientists from the international community in various forms is necessary to help Africans avoid the full undesirable effects of climate change.

## 1. Introduction

Apparent since the early 1970s, climate change and its effects are already a reality in Africa [1] and it has resulted in new and varied phenomena, including increased temperatures, low agricultural production, severe variations in weather patterns and disease transmission, among others. Research indicates that although Africa’s CO_2_ emissions is much lower than the rest of the world, the continent is the most vulnerable to the effects of climate change [2] (Figure 1A). Africa’s vulnerability to climate change hinges on a range of factors, including weak adaptive capacity, high dependence on ecosystem goods for livelihoods and traditional agricultural systems [3]. Vulnerability will be further exacerbated by growing food demand from an increasingly wealthy populations and the reduction of their rural population (Figure 1B–D).

In particular, the consequences of climate change are projected to severely negatively impact agricultural production, food security, water resources, health, energy and ecosystem services, with related effects on lives and sustainable development prospects in Africa [2]. 

Climate change has increased temperatures across the continent and generated, in some cases, extended heat waves [4]. Future projections using general circulation models (GCMs), Representative Concentration Pathway 2.6 (RCP2.6), suggest a temperature increase of 1.7 °C by 2030 [1]. Figure 2 shows historic and predicted temperature and precipitation data corresponding to the African continent. The green thick line, in both plots, represents historic data, from 1991 to 2015. The thin red lines which represent multiple climate models under RCP2.6 suggest a most likely increase in temperature, drought and flood (Source: World Bank, Climate Change Knowledge Portal). Significant increases in temperature, higher than the global mean temperature, are expected, particularly in the Sahel region. Also, regions in Africa within 15 degrees of the equator are projected to experience an increase in hot nights, as well as longer and more frequent heat waves. Projections show that the western Sahel region will experience the strongest drying, with a significant increase in the maximum length of dry spells. Currently, it is one of the most environmentally degraded in the world, with temperature increases projected to be 1.5 times higher than in the rest of the world.

Even though climate change is not at its peak in Africa, past variations in temperature and precipitation have already shown immediate implications for food production and security across the continent. Drought, heat stress and flooding have exacerbated the already low production of crops and livestock and have contributed to increased rates of malnutrition and poverty, leading to reduced quality of life and health. Food shortages have often caused cross-border migration. Future projections predict areas of maize and beans to experience yield reductions of 12–40% by 2050. In addition, the climate suitability of most major crops is expected to shift as the climate warms [5]. West Africa, for instance, has been identified as a climate-change hotspot, with climate change likely to lessen crop yields and production, with resultant impacts on food security. Reductions of 20% and 32% in maize production as a result of climate change have been projected [6,7,8]. The high impact of the effects of climate change in Africa have been attributed to its geographical position and limited adaptive capacity [9], exacerbated by widespread poverty and low levels of development. Another effect of climate change is the outbreak of internal conflicts driven by the depletion of essential resources, such as potable drinking water. Drought, desertification and scarcity of resources have led to heightened conflicts between crop farmers and cattle herders, and weak governance has led to social breakdowns. 

The African continent is particularly vulnerable to the effects of climate change and the implications of procrastinated action or complete inaction will have long and lasting effects on the continent. Further, considering that the world has become a global village which fosters extreme interconnectivity, there is the potential for a spillover of the effects to the rest of the globe. In view of this, the need for all stakeholders, especially African scientists, to prepare and adopt effective and efficient strategies to mitigate these effects cannot be overemphasised. Technologies targeting increased agricultural production exist across the world, some examples are: next generation (Next-Gen) artificial intelligence tools, advances in genomics and phenomics [10,11], the availability of computational resources on plant genomics, advances in plant phenotyping, modelling and computational biology, advances in plant breeding, and improved applications of synthetic biology [12], which are providing insights into complicated biological mechanisms underlying plant response to environmental stresses resulting from climate change while simultaneously providing opportunities to expand the range of possible plants products. It is critical that Africa makes use of current available technologies (e.g., equipment, methods or processes) which will increase the speed at which most agri-food stakeholders can prepare and adapt to the effects of climate change. In addition, it is imperative that breeding programs in Africa are focused on breeding crops for high yield and superior adaptability to new and evolving climates to ensure sustained food security, biomass production and ecosystem services [10]. 

Despite the availability of these technologies, most of them are expensive to acquire, difficult to use, disseminated in pay-for-access journals and mostly tested under specific environments which may be alien to the African environment. Thus, it is also critical that the global scientific community collaborate to support African scientists in the development of basic and applied research. It is only through concerted efforts that African scientists can fully explore ways to create effective and efficient agricultural strategies that are more resilient to climate change and fine-tuned for a rapidly growing population.

In this review, we discuss the current advanced technologies in fields such as genomics and phenomics, and show how these technologies can be utilised to enhance knowledge discovery to mitigate the effects of climate change and increase production in Africa.

## 2. Resources in Plant Breeding

Plant breeding refers to manipulation of plant species in order to create desired genotypes and phenotypes for specific purposes. Plant breeding activities date back to the dawn of agriculture when healthy and strong looking plants were transplanted into other locations with the view to changing the hereditary material of plants. Breeding at the time was long, tedious and rudimentary and yet it yielded results; the best plants in the field were selected and their seeds were kept as stock for future planting. This process led to genetic mutations and sometimes resulted in new useful traits which were bred into economically important crops by human selection [13]. 

Between the end of the 19th century and the beginning of the 20th century, Mendel’s discovery of the laws of inheritance, the discovery of the structure of DNA and the establishment of quantitative genetics theory, plant breeding activities accelerated. Advances in plant breeding during this time included replicated field trials, controlled crossings, statistical analyses, formal experimental designs, hybrid breeding, pedigree-based estimates of breeding values, and precise measurement of yield at scale [14]. In the late 1980s, the advent of complete genetic linkage maps stimulated interest in the systematic genetic dissection of discrete Mendelian factors underlying quantitative traits in experimental organisms [15]. The introduction of high-throughput genotyping expanded the quantitative genetics tools to dissect variation in natural populations [16]. In addition, genome-wide association studies have increased the ability to identify individual variants associated with useful effects in crops. Genomic selection (GS) uses the information contained in dense genetic marker sets for the prediction of quantitative traits. Also, GS lends itself to easy application of computational models derived from artificial intelligence tools to facilitate accurate prediction of phenotypes in a breeding programme [17]. Further, advances in semiconductors and graphics processing unit (GPU) technologies has made it easy for GS to employ deep learning tools to analyse multiple traits with mixed phenotype and prediction accuracy [18].

The commonly adopted conventional breeding methods for crops remain a herculean task. The need for frequent crossing and selfing for the desired number of generations, preceded by extended crop cycle generation during breeding activity, present serious challenges. Currently, speed breeding, a new breeding technique, which uses supplemental light-emitting diode (LED) lighting to extend photoperiod and shorten generation cycles in the greenhouse, has been used to demonstrate the potential to shorten generation cycles of key staple crops such as wheat (*Triticum aestivum*), chickpea *Cicer arietinum*) and oilseed rape (*Brassica napus*). For instance, for chickpea (*Cicer arietinum*) and oilseed rape (*Brassica napus*), it has been demonstrated that six and four generations, respectively, can be obtained per year [14]. Speed breeding is an important step forward, especially for countries which do not have the infrastructure or budget to run several or multiple trials. 

Further, traditional methods to enhance genetic and phenotypic variation in crops has relied on inter-crossing with wild relatives to introduce “exotic” allelic diversity, creating novel alleles by random mutagenesis, and genetic engineering. These approaches are not always guaranteed to produce agronomically meaningful traits [19]. Conversely, targeted mutagenesis technology, which combines clustered regularly interspaced short palindromic repeats (CRISPR) and CRISPR-associated protein 9 (Cas9), popularly recognised as CRISPR/Cas9, can generate desirable mutations by inducing precise gene editing through efficient double strand DNA breaks (DSBs) at a target site. CRISPR-Cas9-based multiplexed gene editing (MGE) provides a powerful method to modify multiple genomic regions simultaneously controlling different agronomic traits in crops [20,21]. CRISPR/Cas9 is now widely used in plants and has made genome editing experiments more efficient, precise and rapid. Akin to CRISPR-Cas9, genome wide association studies (GWAS) are also emerging as powerful tools for the understanding of the inheritance of complex traits via utilization of high throughput genotyping technologies and phenotypic assessments of plant collections. In GWAS, identification of significant associations depends largely on factors including genetic marker coverage, number of individuals studied, and linkage disequilibrium (LD) between causative, as well as linked, polymorphisms. Currently, GWAS has, for instance, been employed successfully in hexaploid wheat for identification of quantitative trait loci (QTL) for yield components, abiotic stress resistance, disease resistance and grain quality [22]

CRISPR-Cas9, advanced genomic selection (GS), genome-wide association study (GWAS), and speed breeding constitute some of the many technological breakthroughs that promise to boost modern breeding.

## 3. Advances in Genomic Selection

The global human population is expected to grow by 25% in the next 30 years, reaching 10 billion. While traditional breeding methods have been beneficial over the past decades, they will outlive their usefulness as they will unable to match the pace required to meet the demand for crops such as wheat (*Triticum aestivum*), rice (*Oryza sativa*) and maize (*Zea mays*) [23,24]. In recent years, the global climate has changed, resulting in drastic fluctuations in rainfall patterns and increasing temperature. Abrupt climate changes can cause significant economic losses to countries worldwide. Breeders, and plant scientists alike, bear the burden of improving the quality of existing crops, as well as generating new ones that possess higher nutrition content, higher yields, and disease and pest-resistance, as well as climate-smart capabilities. 

The introduction of second- and third-generation sequencing platforms means that breeders can afford to use DNA markers to assist selections. This has facilitated gene discovery, trait dissection and predictive breeding technology [25]. Over two decades ago, molecular marker technology was predicted to be a significant tool that would reform breeding programs and facilitate swift gains from selection [26,27]. Currently, however, it appears marker-assisted selection (MAS) has failed to significantly improve polygenic traits [28,29]. While MAS has been effective for the manipulation of large effect alleles with known association to a marker [30], it has been a bottleneck when many alleles of small effect segregate and no substantial, reliable effects can be identified [31]. 

Originally developed for breeding livestock, GS utilises simulated data and is useful for marker-based prediction of breeding values for individual animals [32]. Application of the GS approach promises avenues to mitigate the challenges experienced when MAS is adopted for the identification of quantitative traits [33]. GS primarily focuses on determination of the genetic potential of individual plants instead of finding the specific QTL. The authors of [34] report that preliminary studies of the application of GS to dairy cattle demonstrated significant and enhanced selection accuracy. Further studies indicate that GS outperforms MAS under the same economic investments, even at low accuracies [35,36]. Pioneering studies on the prospects of GS in plant breeding, carried out in maize (*Zea mays* L.), can be attributed to by Bernardo et al. [36], who employed simulated data for their studies. Similarly, studies have been carried out on wheat (*Triticum aestivum* L.) [37], barley (*Hordeum vulgare* L.) [38] and oat (*Avena sativa* L.), with much success [39]. Studies, this approach has been employed to investigate hybrid breeding [40,41] and inbred or double haploid (DH) lines. In all cases the accuracy of predictions demonstrate that GS is more efficient than PS. It is important to note that, prior to the establishment and adoption of GS, plant breeders had developed ideas with similar ingredients to Meuwissen et al.’s GS [32].

Currently, industries within the private sector are employing genomic selection in their maize breeding programmes [42]. Application of this method has produced varieties such as the ‘AQUAmax’ hybrids now widely planted in the United States. According to Cooper et al. [42] and Gaffney et al. [43], AQUAmax maize hybrids demonstrate considerably greater yields under both favourable and unfavourable (e.g., drought stress) conditions. This result establishes enhanced yield stability and reduced risks for maize producers [43].

Using genomic selection, numerous traits can be targeted concurrently in order to increase desired benefits. For instance, accurate selection of phenotypes with measurable multiple traits, including normalised difference vegetation index, canopy temperature and genomic breeding values (GEBV) for increased yield, can be achieved using this method [44]. Utilization of end-use quality traits commonly measured in wheat breeding programs is another example worth mentioning. Here, predictions obtained from nuclear magnetic resonance spectral and near-infrared analyses of small amounts of flour can be combined with DNA marker predictions to provide accurate GEBV which will further facilitate segregation of plants with desired end-use quality traits at an earlier time within the breeding cycle [45].

Benefits obtained from employing genomic selection are enhanced even more when combined with other modern technologies which reduce intervals in generation and identify, as well as integrate, the exact position of causative mutations influencing the desired trait or traits. This is because, in cases like this, predictions will not depend on linkage disequilibrium between the DNA markers and the causative mutations.

Considering that speed breeding has the potential to significantly decreased generation intervals [14], genetic gains from employing this method would be greatly enhanced by coupling it with genomic selection. To achieve this, genomic selection has to be applied at each generation for accurate selection of parents for the next generation. Currently, the high cost of genotyping is the biggest challenge to the application of GS. One option to mitigate this would be to apply the procedure at every second or third generation with the view to selecting candidates which meet the required benchmarks for traits that can be phenotyped with a high level of reliability and confidence during speed breeding cycles [46].

## 4. Advances in Plant Phenotyping

It has been two decades since Schilling et al. [47] proposed the idea of phenomics as the discipline that analyses, interprets, and models the genotype–phenotype relationship. Schilling et al. [47] also proposed the use of in silico models that utilise genomics data to predict phenotypic traits, such as the visual appearance of an organism and its metabolic response to a given stimulus. However, at the time of Schilling’s paper advances in sensor technologies and mechatronics lagged behind. Likewise, robust algorithms for the integration, analysis and visualisation of multidimensional data were none existent. As a result, the phenomics idea failed to fully materialise. 

Fast forward 11 years, Houle et al. [48] faced a different technological environment. New multi-processor/multi-core technologies [49] gave way to the construction of more powerful phenotyping technologies to better understand the genotype–environment relationship [50]. Such phenotyping technologies can track the formation and development of static and dynamic traits at the cellular and plant levels respectively [51,52]. For example, Camargo et al. [53] phenotyped a cohort of the wheat MAGIC population across their life cycle and under controlled conditions. Assessment of temporal dynamics of plant height, area and senescence allowed the identification of marker-trait associations and tracking of trait development against the genetic contribution of key markers. Wang et al. [54] used unmanned aerial vehicle high-throughput phenotypic platforms (UAV-HTPPs) to measure the height of maize inbred lines at four growth stages. QTL mapping identifies multiple loci controlling plant height at different growth stages with a small number of them being common across all growth stages. As such, plant phenomics offer a suite of technologies to accelerate progress in understanding gene function and environmental responses. Wang et al. [55] used confocal imaging and kinetic root elongation assays to analyze the dynamic progress of ethylene-induced microtubule reorientation in Arabidopsis thaliana. They demonstrated that the time courses of ethylene-induced microtubule reorientation and root elongation inhibition are highly correlated, and that microtubule reorientation is required for the full responsiveness of root elongation to ethylene treatment. 

## 5. Modelling and Computational Biology Applications

Computational modelling uses mathematical and statistical principles to describe biological processes. Models are usually accompanied by data visualization strategies to enable scientists to examine the data provided and subsequently model phenomena in a more holistic manner, as well as facilitate accurate interpretation of the model and related outcomes. As such, computational models are used agricultural research to explore the development of complex organisms and how they respond to given circumstances such as stresses caused by endogenous and exogenous factors. These virtual observations and predictions can facilitate the development of crop ideotypes designed to meet future yield and nutritional demands [56].

Computational models are usually dynamic, they capture and model spatial–temporal responses which in the case of complex trait analysis can have the potential to extend simulations out to novel environments and lend mechanistic insight to observed phenotypes. Typical examples include utilization of near-infrared (NIR) measurements to model protein and moisture content in harvested wheat in Australia [57,58,59] and also to identify molecular differences in cereal (barley) mutants [59,60]. Recently, commercial entities like Bayer have employed modelling and computer simulation to reduce cross-breeding steps required to obtain required characteristics in plants. In addition, their experiments have demonstrated the possibility of using less plants than previously needed (Bayer Research, 2016). Despite the translational opportunities for varietal crop improvement that could be unlocked by linking natural genetic variation to first-principles based modelling, these models are challenging to apply to large populations of related individuals [61].

Furthermore, advances in synthetic biology have already demonstrated the capacity to design artificial biological pathways whose behaviour can be predicted and controlled in microbial systems. For instance, desirable characteristics such as nutritional value and increased crop yield can be achieved using well-engineered plant specific synthetic metabolic pathways. Despite the demonstrated potential, challenges including the inability to characterise plant cellular pathways, as well as complexity arising as a result of compartmentalization and multicellularity, present limitations to the technology. Increasing developments in modern computational capabilities is providing much needed solutions to these challenges by making available the means to test the feasibility of plant synthetic metabolic pathways, despite gaps in the accumulated knowledge of plant metabolism [62].

## 6. Synthetic Biology

The United Nations estimates that the world population will reach 10 billion by 2057. A population of such magnitude requires an assured and steady supply of basic food that guarantees socioeconomic stability across the world. This means, to meet food demands of this growing population, crop yield has to double in the next 30 years, representing an annual yield increase of 2.2% [3,6]. Currently, development and global distribution of natural and synthetically generated fertilisers (particularly nitrogen, phosphorus and potassium), coupled with the green revolution, which has encouraged utilization of non-traditional breeding methods to maximise plant architecture and light harvesting, have increased production of a number of staple foods. However, these strategies are no longer sufficient to guarantee food production under the current social and environmental challenges, such as the loss of arable land to rapid urbanisation, climate change and the fast emergence and re-emergence of plant diseases. To mitigate this situation, there is the need to devise novel and innovate methods to enhance agricultural productivity, with a view to guaranteeing food security in the medium and long term. Genetically engineering plant performance towards improving growth and yield is a potential solution to overcome upcoming problems [7,8]. 

Synthetic biology, a branch of biotechnology, is emerging as the key strategy to enhance agricultural productivity and guarantee food security. Research in this field uses a set of approaches and tools within biotechnology to enable modification or creation of biological organisms. Synthetic biology involves the application of engineering principles and advances in molecular, cell, and systems biology to describe and understand/recreate core biological processes. For instance, one focus of synthetic biology is the design and construction of core bio components, from enzymes to genetic circuits and metabolic pathways that can be modelled, understood and tuned to fulfil a specific role. The assembly of these smaller parts and devices into larger integrated systems can be used to solve specific problems such as the optimisation of nitrogen fixation or CO_2_ sequestration by plants and microorganisms. A model example of synthetic biology is synthetic nucleases. Fusion of synthetic domains modeled on the DNA-binding domains of zinc finger (ZNF) proteins or transcription activator-like effectors (TALEs) with the nuclease domain of a natural restriction endonuclease, synthetic enzymes have been developed that enable scientists to induce double-strand breaks (DSBs) in any genomic locus they wish [63].

Modular genetic fragments constitute the key components which are combined to create synthetic biological systems. For full functionality, each part requires its own unique mathematical model-guided designs and quantitative function characterisation [64]. Currently, plant synthetic biology still lags behind yeast, bacterial and mammalian systems, which have significantly contributed to research efforts and are also extensively applied in the biotechnological and biopharmaceutical industries [65,66]. Standardization of genetic parts and development of modular cloning tools constituted the pioneering efforts towards establishing a more generalised implementation of synthetic biology strategies for plant research and applications [67,68]. Reported potential benefits of applications of synthetic biology to efforts to increase food production and quality include development of synthetic metabolic routes for improved CO_2_ fixation and carbon-conservation, reduction of natural and synthetic fertiliser application in agriculture by engineering nitrogen fixation in crop plants and construction of synthetic plant microbiome consortia, increased nutritional value of crop plants and the utilisation of photoautotrophic organisms as production platforms for commercially interesting compounds [69].

Given the importance of synthetic biology, groups such as Open Plant (https://www.openplant.org/) and the Joint BioEnergy Institute (https://public-registry.jbei.org/) are currently developing open-source registries for plant-specific DNA parts [12]. Also, Synbio Africa, a forum for researchers, students, citizen scientists, policy makers and the public at large, convenes to strategise and develop successful pathways for the propagation of synthetic biology technologies, products and services throughout Africa [70]. Again, there is the Biomaker Africa programme which aims to train biologists, as well as non-biologists, to learn, design, prototype and share science hardware critical to building tools for laboratory use and environmental sensing.

## 7. Computational Resources on Plant Genomics

Advances in high-throughput genomics technologies have greatly accelerated the progress in both fundamental plant science and applied breeding research. In recent years, the genomic sequences of numerous plant species, including common crops such as wheat, have been sequenced [71]. Computational tools have been developed to deal with the questions of which plant has been sequenced and where is the sequence hosted. Although, short-read sequencing technologies have made genome sequencing faster and more affordable, closing genomes is often costly and assembling short reads from draft genomes whose fragmented into many contigs remains a challenge. Thus, long-read, single-molecule sequencing is a relatively new sequencing technology that is designed to overcome some of the deficiencies of short-read sequencing. The PacBio platform [72] and the Oxford Nanopore Technologies (ONT) MinION [73] are both long-read, single-molecule sequencing technologies. The MinION Oxford’s Nanopore is a small hand held sequencing device, which is based on nanopores. MinION can be plugged directly into a laptop via a USB3 port and requires a relatively small upfront financial investment relative to PacBio. This affordability and simplicity has enabled the rapid uptake of MinION sequencing by individual labs worldwide and facilitated new applications such as the rapid on-field detection of Cassava mosaic begomoviruses in Sub-Saharan Africa [74] or the assembly of highly contiguous genome of *A. brassicae* [75] in India. Despite the rapid evolution and constant improvement of next generation sequencing technologies, many standard pipelines and tools that can potentially assemble a reasonable quality genome are available for both sequencing technologies [76]. What is required is for the user to have an acceptable level of familiarity with the technology, the data and the genome to be assembled. All users are also required to have an acceptable level of proficiency with data analytics and visualisations. It is usually the case that most genome assemble tools are developed under the Python or R environment. Some tools are available in graphic user interface (GUI) form, they are usually easy to use and allow users to upload data into the cloud to perform some analytics. The CyVerse platform [77], an US National Science Foundation, provides life scientists with powerful computational infrastructure to handle huge datasets and complex analyses which enables data-driven discovery. CyVerse also provides data storage, bioinformatics tools, image analyses, cloud services and APIs. The only limitation of CyVerse is that it requires users to upload their data on the cloud. This is big limitation when it comes to assemble even small genomes. Thus, a combination of GUI and scripts tools is perhaps the best approach to perform reasonably-sized genome assemblies.

## 8. Challenges and the Way Forward 

Application of emerging technologies have the potential to tackle some of Africa’s urgent nutritional and environmental demands. Although some of the approaches mentioned in this review are at least partially known to African scientists, most scientific discoveries remain unknown. African scientists are still very isolated from the global scientific community, but new policies on Open Access publication are becoming a vital strategy to disseminate scientific knowledge across Africa. Open access not only allows African scientists to stay up to date with newly available methods and technologies but also to identify experts in specific fields. The downside of Open Access is the high publication costs, which seem irrelevant to scientists is most developed countries but are an important issue for African scientists, as the costs are prohibitively high. Thus, Open Access policies should be revisited and for example put pressure on open access journals to waive charges for researchers in developing countries or to encourage academics to write first for journals that are affiliated to societies. Profits from these kinds of journals go back into supporting science through research grants, travel grants and meeting support. Another way to stay up to date with science is through social media, these virtual channels encourage and facilitate scientific exchange and collaboration. It is also very important for scientists to have a web presence to facilitate and initiated research collaborations.

Assuming that the knowledge gap can be narrowed down, the next big barrier is concerned with technology transfer. In Africa, most agri-related training is coordinated by the Consultative Group for International Agricultural Research (CGIAR) organizations. Thus, scientists have to travel to or from CGIAR’s headquarters to receive or provide training. As a result, training is not regular or widely available. Alternative strategies such as the use of voice over IP (VoIP) technologies to facilitate training and collaboration should be explored. Another strategy is to use a multiplicative approach to training whereby an expert trains a cohort of students and those students are then expected to train another group. Appropriate trainings strategies should be in place to guarantee the comprehension newly acquired knowledge.

In addition to knowledge and training, scientists also need to be able to work in environments that allow them to develop their own research and to make significant contributions to their field of expertise. Therefore, African governments need to develop strategies that lure scientists to stay and further scientific knowledge in Africa rather than go to other countries that offer more opportunities for research and development. Giving African scientists the tools and resources to develop their research is the best strategy to prepare Africa for the most likely ravaging effects of climate change.

Though farmers may not directly apply the emerging technologies reviewed here, they form the core of direct beneficiaries of processes, methods and products which emanate from research efforts. They are also catalysts for adoption of new and improved crop varieties for domestic and commercial production and consumption. For instance, farmers, through experience and application of their indigenous methods, are able to identify crop characteristics. Again, their constant interaction with retailers and consumers provide them with a wealth of information critical to crop production and improvement activities. It would good for scientists to identify ways to engage farmers in knowledge exchange in order to accurately capture and develop the farmer- and consumer-preferred varieties required. Recently, agri-related NGOs are gaining attention. These organisations utilise their influence and adopt advocacy to create changes and change paradigms. Empowered with information and exposure to these technologies, they could blaze the trail of education, facilitating for communities the adoption of improved methods and varieties for increased production.

## 9. Conclusions

Presently, climate change is already taking a huge toll on the African continent, with evidence particularly in the production of crops. Projections suggest the effect of climate change could be worse than originally anticipated. Added to that, are the current challenges associated to food security in Africa, such as population growth and the decrease of the rural population. Thus, the African continent needs to adopt implement measurements, such as the adoption of effective and efficient technological strategies to mitigate the impact of climate change and ensure food security in the short and long term. 

This work has identified and reviewed current and emerging technologies which possess the potential to mitigate climate change-related challenges likely to negatively impact agricultural production, nutritional and environmental demands in Africa. Emerging technologies which would be useful include next generation (Next-Gen) artificial intelligence tools, advances in genomics and phenomics, availability of computational resources on plant genomics, advances in plant phenotyping, modelling and computational biology, advances in plant breeding, and improved applications of synthetic biology. 

In addition to the urgency for the adoption of these technologies, significant challenges exist for successful adoption and sustained use. Challenges identified by this work include high upfront cost and utilization of the technologies, lack of capacity, lack of technology transfer infrastructure and strategies, needed legislature in some cases. 

This work has also identified the use of descriptive and predictive models driven by data as critical to enhancing the capacity of African research scientists. Against the background of the high cost of relevant software, lack of access to high performance computers and low capacity of scientists to effectively apply the software for modelling purposes, this work recommends improvement capacity of African scientists in areas including software training and artificial intelligence while tackling the issue of training cost and related risks. 

At this point, we suggest technological adoption rather than complete technological creation, for three reasons. Firstly, development of technologies from scratch may not be the feasible and effective option due to the imminent change in climate and the vulnerability of Africa as a whole. Again, the time required to develop, test and deploy new and working technologies is generally longer than it is often anticipated and new technologies may not be ready for use at the time required. Finally, access to expert knowledge derived from already tested and deployed technologies is currently available. 

The work recommends collaborations between African scientists and their counterparts to enable effective technology transfer. Previously, scientific conferences and scientific missions were the main platforms for scientific exchange and collaboration. Recently, social media and other internet platforms have become major drivers in the democratization of knowledge, including scientific knowledge. African scientists can also use social media platforms to gain strong scientific presence and facilitate the initiation of scientific collaborations. Governments of Africa are also encouraged to provide the tools and incentives which will provide African scientists with the stability to present meaningful significant and applicable research findings which can easily be adopted to mitigate the looming effects of climate change. 

As part of future work, the authors plan to test and report on the application of two technologies, genetic mapping and speed breeding on selected crops in two developing countries. 

## Figures and Tables

**Figure 1 plants-09-00381-f001:**
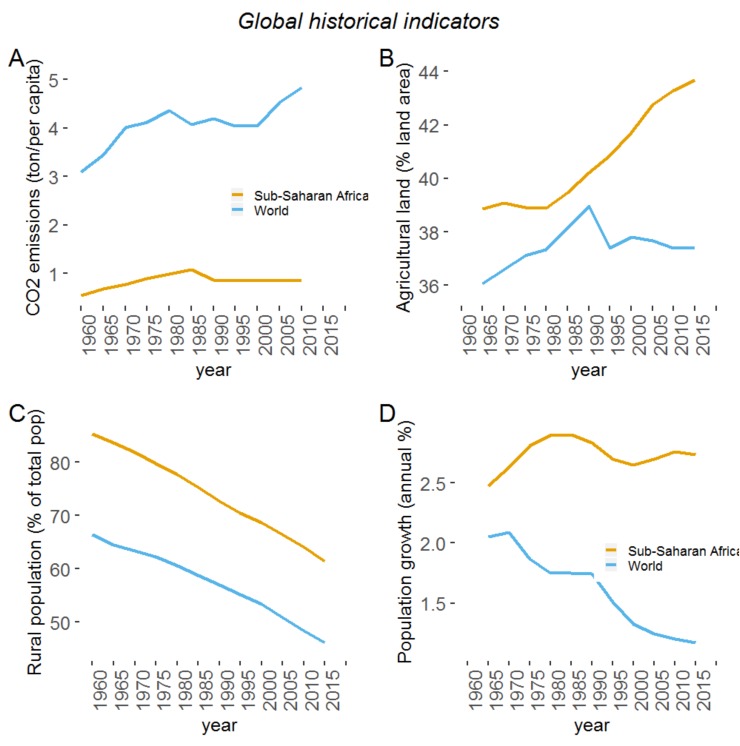
Comparison of global historical indicators (1960–2015) between Africa and the rest of the world for: (**A**) CO_2_ emissions, (**B**) agricultural land, (**C**) rural population and (**D**) population growth. (Source: World Bank, 2020).

**Figure 2 plants-09-00381-f002:**
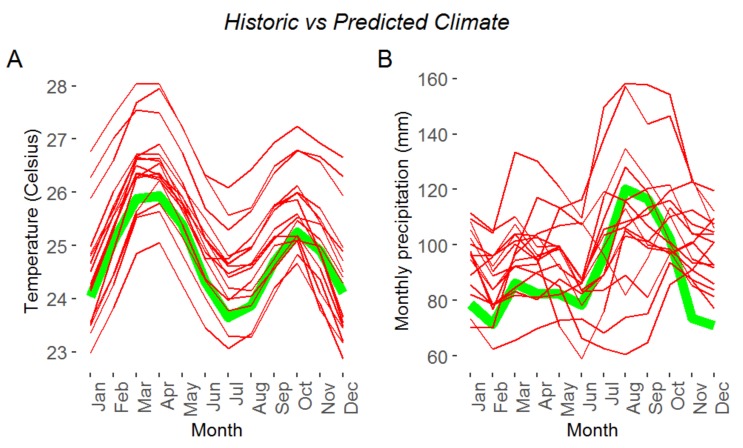
Historic and predicted (2020–2039, RCP 2.6) (**A**) temperature and (**B**) precipitation. The green tick line shows average monthly data from 1991 to 2016. Thin red lines are the outcome of multiple models predicting average monthly data from 2020 to 2019 under Representative Concentration Pathway 2.6 (RCP2.6). RCP2.6 assumes that global annual greenhouse gases (GHG) emissions (measured in CO_2_-equivalents) peak between 2010–2020, with emissions declining substantially thereafter. (Source: World Bank).

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
