# Peer review of "Emerging Advanced Technologies to Mitigate the Impact of Climate Change in Africa"

_plants, 2020, doi:10.3390/plants9030381_

Round 1

Reviewer 1 Report

1, line 227, there is the need to "device" novel and innovate methods to enhance, it should be devise 2, please provide more review on specific case studies in each technologies, so it is more straightforward to see what is the status now and how promising it is to adopted.

Author Response

Reviewer 1

  1. line 227, there is the need to "device" novel and innovate methods to enhance, it should be “devise”

The sentence has been corrected and highlighted in the manuscript.

  1. please provide more review on specific case studies in each technology, so it is more straightforward to see what is the status now and how promising it is to adopted.

Specified case studies have been presented during the discussion of each technology as recommended.

Reviewer 2 Report

In the article, both in the abstract and in the introduction, the purpose of the study was not clearly stated. There is only information about emerging technologies and their adaptation, while one could expect a clearly stated goal set by the authors of the reviewed material.

Generally, the article includes many considerations on advance(s) in plant production, so maybe the word “advance” can be included in the title of the manuscript ?

Developing the issue of development and implementation of new technologies, it would be worth at the beginning of the article to define the concept of “technology” to make it easier to refer to what technology covers. This way it will be more clear for the reader to recognize scope of the undertaken considerations.

Lines 78-79 include the name / acronym GPU technologies. It is necessary to give full name of the GPU acronym, especially when it is used first time in the text. The same note applies to CRISPR / Cas9. The article uses two forms, i.e. CRISPR / Cas9 and CRISPR-Cas9. Do these two acronyms mean the same thing? The acronym GWAS was not developed in the work, too.

In the Conclusion chapter there is included ICT acronym without the description. There should be given full name of the ICT acronym.

Line 93 includes the form “[word missing]”. What is it ?

Line 105: the Latin names of plants should be given in “Italic” type (font).

The article says that CRISPR-Cas9, GS, GWAS, and speed breeding constitute some of the many technological breakthroughs, so it can be valuable to write something about these technologies.

Some materials given in the References should be formatted according to the Editor rules including for example place, where year is provided.

It is written in the paper about scientists as a recipient of knowledge and especially the advance concerning emerging technologies. But in my opinion it can be valuable or more valuable to write about other group of recipient of knowledge and emerging technologies, i.e. farmers and  / or managers working in African agriculture / food economy. There should be written, what is directly offered by emerging technologies for peoples directly engaged in plant production in the farms. The Conclusion chapter and some suggestions on further development include first of all scientists, eg. “… to increase the capacity of African scientists …”, “It also recommends collaborations between African scientists and …”, “Finally, governments of Africa are encouraged to provide the tools and incentives which will provide African scientists with …”. Please, write something about association between emerging technologies and farmers.

The Conclusion chapter includes the sentence “This work recommends improvement in provision of ICT as a training tool …” but I can not understand what is expressed by the ICT as a training tool. There should be given more details.

Is it possible to make the paper more scientific ? Some data, their description and analysis can help to reach this aim. Title of the paper includes climate change in Africa, so it can be nice to write more details about previous, current and expected changes in African climate including changes in temperature, microclimatic conditions, pollution and other ones. This way it will be more justified to present scientific problem of application of emerging technologies as challenge for the expected changes in African climate.

Author Response

Reviewer 2

  1. In the article, both in the abstract and in the introduction, the purpose of the study was not clearly stated. There is only information about emerging technologies and their adaptation, while one could expect a clearly stated goal set by the authors of the reviewed material.

The purpose of the study was to identify and present emerging advanced technologies which possess the potential to mitigate the impact of climate change in Africa but are currently underutilized or not available on the continent. The purpose has been integrated and highlighted in the manuscript.

  1. Generally, the article includes many considerations on advance(s) in plant production, so maybe the word “advance” can be included in the title of the manuscript?

The authors are grateful to the reviewer for the suggestion. The suggestion has been implemented. The title now reads: “Emerging advanced technologies to mitigate the impact of climate change in Africa”. This has been highlighted in the manuscript.

  1. Developing the issue of development and implementation of new technologies, it would be worth at the beginning of the article to define the concept of “technology” to make it easier to refer to what technology covers. This way it will be clearer for the reader to recognize scope of the undertaken considerations.

The scope of the technology for this work has been defined on page 1 and highlighted in the manuscript.

  1. Lines 78-79 include the name / acronym GPU technologies. It is necessary to give full name of the GPU acronym, especially when it is used first time in the text. The same note applies to CRISPR / Cas9. The article uses two forms, i.e. CRISPR / Cas9 and CRISPR-Cas9. Do these two acronyms mean the same thing? The acronym GWAS was not developed in the work, too.

GPU stands for graphics processing unit. Also, clustered regularly interspaced short palindromic repeats (CRISPR) and CRISPR-associated protein 9 (Cas9) popularly known as CRISPR-Cas9 are two separate components which are usually combined to facilitate addition, removal of genetic material or make alterations at desired locations in the genome. Further, genome wide association studies (GWAS) has been defined and briefly discussed in the manuscript.

  1. In the Conclusion chapter there is included ICT acronym without the description. There should be given full name of the ICT acronym.

The full name of the acronym has been provided and highlighted in the manuscript.

  1. Line 93 includes the form “[word missing]”. What is it ?

The anomaly has been corrected.

  1. Line 105: the Latin names of plants should be given in “Italic” type (font).

The Latin names of the plants have been provided and highlighted in the manuscript.

  1. The article says that CRISPR-Cas9, GS, GWAS, and speed breeding constitute some of the many technological breakthroughs, so it can be valuable to write something about these technologies.

A brief description of the afore-mentioned technologies has been provided and highlighted in the “Resources in Plant Breeding” section of the manuscript.

  1. Some materials given in the References should be formatted according to the Editor rules including for example place, where year is provided.

All references have been formatted accordingly to journal requirements.

  1. It is written in the paper about scientists as a recipient of knowledge and especially the advance concerning emerging technologies. But in my opinion it can be valuable or more valuable to write about other group of recipient of knowledge and emerging technologies, i.e. farmers and / or managers working in African agriculture / food economy. There should be written, what is directly offered by emerging technologies for peoples directly engaged in plant production in the farms. The Conclusion chapter and some suggestions on further development include first of all scientists, eg. “… to increase the capacity of African scientists …”, “It also recommends collaborations between African scientists and …”, “Finally, governments of Africa are encouraged to provide the tools and incentives which will provide African scientists with …”. Please, write something about association between emerging technologies and farmers.

The authors are grateful for the comment. The role of famers and other stakeholders like NGOs have been integrated and highlighted in the manuscript.

  1. The Conclusion chapter includes the sentence “This work recommends improvement in provision of ICT as a training tool …” but I cannot understand what is expressed by the ICT as a training tool. There should be given more details.

The sentence has been modified and highlighted in the manuscript.

  1. Is it possible to make the paper more scientific ? Some data, their description and analysis can help to reach this aim. Title of the paper includes climate change in Africa, so it can be nice to write more details about previous, current and expected changes in African climate including changes in temperature, microclimatic conditions, pollution and other ones. This way it will be more justified to present scientific problem of application of emerging technologies as challenge for the expected changes in African climate.

A section which discusses climate change and its impact on the African continent has been integrated into the introduction and highlighted. Additional information on current projections of the effects of climate change has also been presented. We also added two plots, Figure 1 shows historical socio-economic indicators between Africa and the world and Figure 2 shows historic and predicted climate

Reviewer 3 Report

Review work prepared in a diligent and accurate manner. The possibilities of using the current technologies in fields such as genomics and phenomics and show how these technologies can be utilized to enhance knowledge discovery in Africa are described.
The publication summarizes a lot of interesting information. however, the reviewer expects to make recommendations on climate change based on a literature review and own research as part of such work. There are no specific conclusions, and the part entitled Conclusions is a simple summary. Please complete the publication with konktrene conclusions, recommendations and information on what else needs to be done regarding climate change. Please assign tasks for the next stages of research on the topic described in the publication.

Author Response

Review work prepared in a diligent and accurate manner. The possibilities of using the current technologies in fields such as genomics and phenomics and show how these technologies can be utilized to enhance knowledge discovery in Africa are described.

The publication summarizes a lot of interesting information. however, the reviewer expects to make recommendations on climate change based on a literature review and own research as part of such work. There are no specific conclusions, and the part entitled Conclusions is a simple summary. Please complete the publication with konktrene conclusions, recommendations and information on what else needs to be done regarding climate change. Please assign tasks for the next stages of research on the topic described in the publication.

Conclusions and recommendations have been provided as recommended. Further, the next steps regarding climate change has been provided. Finally, the next tasks for the research on the topic have also been presented.

Round 2

Reviewer 1 Report

no more comments.

This manuscript is a resubmission of an earlier submission. The following is a list of the peer review reports and author responses from that submission.